# Neural Decoding of Visual Imagery via Hierarchical Variational Autoencoders

## Abstract

Reconstructing natural images from fMRI recordings is a challenging task of great importance in neuroscience. However, current architectures fail to efficiently capture the hierarchical processing of visual information processing, which may bottleneck their representation capacity. Motivated by that fact, we introduce a novel neural network architecture for the problem of neural decoding. Our architecture uses Hierarchical Variational Autoencoders (HVAEs) to learn meaningful representations of natural images and leverages their latent space hierarchy to learn voxel-to-image mappings. By mapping the early stages of the visual pathway to the first set of latent variables and the higher visual cortex areas to the deeper layers in the latent hierarchy, we are able to construct a latent variable neural decoding model that replicates the hierarchical visual information processing. Our model achieves better reconstructions compared to the state of the art and our ablation study indicates that the hierarchical structure of the latent space is responsible for that performance.

## 1 Introduction

Decoding visual imagery from brain recordings is a key problem in neuroscience. This problem aims to reconstruct the visual stimuli from fMRI recordings taken while the subject is viewing the stimuli. Even though some of the excitement is fulled by science fiction and the difficulty of the problem (1), the scientific consensus is that neural decoding has real-world, important implications. It is important for understanding how neural activity relates to external stimuli (2), for engineering application such as brain-computer interfaces (3) and for decoding imagery during sleep (4). Given its importance, neuroscience and machine learning researchers have jointly led the development of sophisticated deep learning architectures that allows us to design pipelines that map voxel-based recordings to the corresponding visual stimuli. Based on the target learning task, visual decoding can be categorized into *stimuli classification*, *stimuli identification*, and *stimuli reconstruction*. The former two tasks aim to predict the object category of the presented stimulus or identify the stimulus from an ensemble of possible stimuli. The reconstruction task, which is the most challenging one and the main focus of this paper, aims to construct a replica of the presented stimulus image from the fMRI recordings.

**Related Work**. The proposed methods for the problem of neural decoding can be broadly classified in three categories: *non-deep learning methods*, *non-generative deep learning methods* and *generative deep learning methods*. The *non-deep learning* class consists of methods that are based on primitive linear models and aim in reconstructing low-level image features (5). Such approaches first extract handcrafted features from real-world images, such as multi-scale image bases (6) or Gabor filters (7), and then learn a linear mapping from the fMRI voxel space to the extracted features. Due to their simplicity, linear models are not able to reconstruct complex real-world images and thus their applicability is restricted to simple images containing only low-level features.

Methods that use convolutional neural networks as well as encoder-decoder architectures belong to the *non-generative deep learning* class. Horikawa et al. (8) demonstrated a homology between human and machine vision by designing an architecture with which the features extracted from convolutional neural networks can be predicted from fMRI signals. Based upon those findings, Shen et al. (1) used a pretrained VGG-19 model to extract hierarchical features from stimuli images and learned a mapping from the fMRI voxels in the low/high area to the corresponding low/high VGG-19 features. Beliy et al. (9) designed a CNN-based Encoder-Decoder architecture, where the encoder learns a

mapping from the stimulus images to the fMRI voxels and the decoder learns the reverse mapping. By stacking the components back-to-back, the authors train their network using self-supervision, thereby addressing the inherent scarcity of fMRI-image pairs. Following up on that work, Gaziv et al. (10) improved the reconstruction quality by training on a perceptual similarity loss function, which is calculated by first extracting multi-layer features from both the original and reconstructed images and comparing the extracted features layer-wise. Such a perceptual loss is known to be highly effective in assessing the image similarity and accounts for many nuances in the human vision (11).

In the *generative deep learning* class, we have model architectures, such as generative adversarial networks (GANs) and variational autoencoders (VAEs). Shen et al. (1) extended their original method to make the reconstructions look more natural by conditioning the reconstructed images to be in the subspace of the images generated by a GAN. A similar GAN-prior was used by Yves et al. in (12), where the authors also introduced unsupervised training on real-world images. Fang et al. (13) leverage the hierarchical structure of the information processing in the visual cortex to propose two decoders, which extract information from the low and high visual cortex areas, respectively. The output of those decoders is used as a conditioning variable in a GAN-based architecture. Shen et al. (14) trained a GAN using a modified loss function that includes an image-space and perceptual loss in addition to the standard adversarial loss. A line of work by Seeliger et al. (15), Mozafari et al. (16) and Qiao et al. (17) assumes that there exists a linear relationship between the brain activity and the GAN latent space. These methods use the GAN as a real-world image prior to ensure that the reconstructed image has some "naturalness" properties. The work by VanRullen et al. (18) and Ren et al. (19) utilize VAE-GANs (20), a hybrid model in which the VAE decoder and GAN generator are combined. In the former work, the authors use the VAE to extract meaningful representations of the data and learn a linear mapping between the latent vector and the fMRI patterns. In the later work, the authors propose a dual-VAE architecture where both the real-world images and fMRI voxels are converted into latent representations, which are then fed as conditioning variable in a GAN. Finally, the work by Lin et al. (21) leverages multi-modality and encodes the fMRI signals into a visual-language latent space and a contrastive loss function to incorporate low-level visual features to the schematic pipeline. Then, the authors use a conditional generative model to reconstruct the images and obtain photo-realistic and accurate reconstructions.

**Contributions**. In this paper, we purpose a novel architecture for the problem of decoding visual imagery from fMRI recordings. Motivated by the fact that the visual pathway in the human brain processes stimuli in a hierarchical manner, we postulate that such a hierarchy can be captured by the latent space of a deep generative model. More specifically, we use Hierarchical Variational Autoencoders (HVAE) (22) to learn meaningful representations of stimuli images and we train an ensemble of deep neural networks to learn mappings from the voxel space to the HVAE latent spaces. Voxels originating from the early stages of the visual pathway (V1, V2, V3) are mapped to the earlier layers of latent variables, whereas the higher visual cortex areas (LOC, PPA, FFA) are mapped to the later stages of the latent hierarchy. Our architecture replicates the natural hierarchy of visual information processing in the latent space of a variational model. Our experimental analysis suggests that hierarchical latent models provide better priors for decoding fMRI signals and, to the best of our knowledge, this is the first approach that uses HVAEs in the context of neural decoding.

## 2 VISUAL INFORMATION PROCESSING

In this section, we give a brief overview of the visual information processing in the human brain and describe the two streams hypothesis, which we use in our experimental architecture. Visual information received from the retina of the eye is interpreted and processed in the visual cortex. The visual cortex is located in the posterior part of the brain, at the occipital lobe, and it is divided into five distinct areas (V1 to V5) depending on the function and structure of the area. Visual stimuli received from the retina travel to the lateral geniculate nucleus (LGN), located near the thalamus. LGN is a multi-layered structure that receives input directly from both retinas and sends axons to the primary visual cortex (V1). V1 is the first and main area of the visual cortex where visual information is received, segmented, and integrated into other regions of the visual cortex. Based on the *two streams hypothesis* (23), following V1, visual stimuli can take the *dorsal pathway* or *ventral pathway*. The dorsal pathway consists of the secondary visual cortex (V2), the third visual cortex (V3), and the fifth visual cortex (V5). The dorsal stream, informally known as the "where" stream, is responsible for visually-guided behaviors and localizing objects in space. The ventral stream, also known as the

"what" stream, consists of V2 and fourth visual cortex (V4) areas and is responsible for processing information for visual recognition and perception. Visual processing occurs hierarchically at three distinct levels (24). The *low-level* includes the retina, lateral geniculate nuclei (LGN), and the primary visual cortex (V1). Low-level processing is the initial step when interpreting an image and it is the place where orientation, edges, and lines are perceived. Sequentially, the *mid-level* processing consists of the secondary (V2), third (V3) and fourth (V4) which extract shapes, objects and colors. Finally, the *high-level* processing consists of category-selective areas such as the fusiform face area (FFA), lateral occipital (LOC), parahippocampal area (PPA) and medial temporal area (MT/V5). These areas show selective response to faces, objects/animals, places and motion, respectively.

Despite the evident hierarchical structure of visual information processing, most current methods for neural decoding fail to fully exploit that fact. Current methods take into account the hierarchy of visual information processing either by mapping the fMRI voxel to hierarchical CNN-extracted image features via regression models (1; 25) or by training an end-to-end DNN model on a feature loss function (12; 14). The major issue with such approaches is that the hierarchy is taken into account in the *feature space* of a CNN model, which is, in general, complex, high-dimensional space. In this work, we propose to take into account the aforementioned hierarchy in the *latent space* of a deep model. Latent spaces are known to produce compact, low-dimensional embeddings of the data and

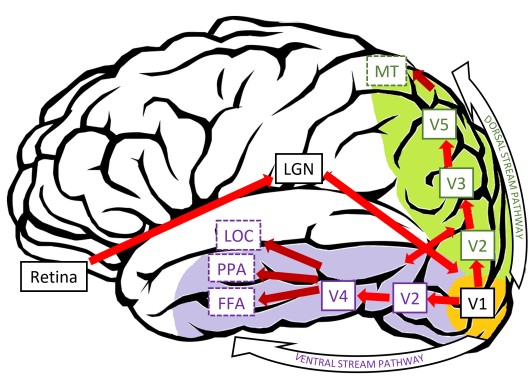

Figure 1: Two stream hypothesis of visual information processing in the human brain.

have recently shown impressive performance on image reconstruction and generation tasks (22). Given these facts, we postulate that a hierarchical latent space provides better priors for decoding fMRI signals. The intuition is that each brain area, being "responsible" for a certain set of features, better be mapped on a compact, low-dimensional representation of those features. For example, given that V1 is broadly responsible for encoding low-level features (e.g., edges, orientations), it is sensible to map the fMRI voxels from the V1 region onto a representation of the underlying images features; and this mapping is much easier to be learned on the latent space, rather than the feature space.

## 3 METHOD

Leveraging the aforementioned intuition, we introduce a neural decoding method that mimics the hierarchical visual information processing in the *latent space*. Our architecture has two main components: a *Hierarchical Variational Autoencoder* (*HVAE*) and a *Neural Decoder*. The HVAE is used for learning compact, hierarchical latent representations of real-world images and is trained using self-supervision. The Neural Decoder is used for mapping the brain signals to the HVAE hierarchical latent space and is trained via supervision on {fMRI,Image} pairs. In this section, we describe each of the components in more detail. Our architecture is visualized in Fig 2 for the special case of 2 latent hierarchical layers.

### 3.1 HIERARCHICAL VARIATIONAL AUTOENCODERS

To capture the inherent hierarchical structure of visual information processing, we propose to model images via a family of probabilistic models known as Hierarchical Variational Autoencoders (HVAEs). HVAEs extend the basic Variational Autoencoder (VAEs) by introducing a hierarchy of latent variables. Formally, let $\mathbf{x}$ be an image and $\mathbf{z} = \{\mathbf{z_1}, \mathbf{z}_2, \ldots, \mathbf{z}_L\}$ be a set of $L$ latent variables. The *generative distribution* or *decoder* is defined as $p_\theta(\mathbf{x}|\mathbf{z}) = p_\theta(\mathbf{x}|\mathbf{z}_1) \prod_{i=1}^{L} p_\theta(\mathbf{z}_{i+1}|\mathbf{z}_i)$ and is parametrized by $\theta$. The prior distribution is defined as $p(\mathbf{z}) = p(\mathbf{z}_1) \prod_{i=1}^{L} p(\mathbf{z}_{i+1}|\mathbf{z}_i)$. The posterior $p(\mathbf{z}|\mathbf{x})$ is approximated by the *variational distribution* or *encoder* $q_\phi(\mathbf{z}|\mathbf{x}) = q_\phi(\mathbf{z}_1|\mathbf{x}) \prod_{i=1}^{L} q_\phi(\mathbf{z}_{i+1}|\mathbf{z}_i)$, which is parametrized by $\phi$. Both the prior and the approximate posterior are represented by factorial Normal distributions. The variational principle provides a tractable lower bound, known as *Evidence*

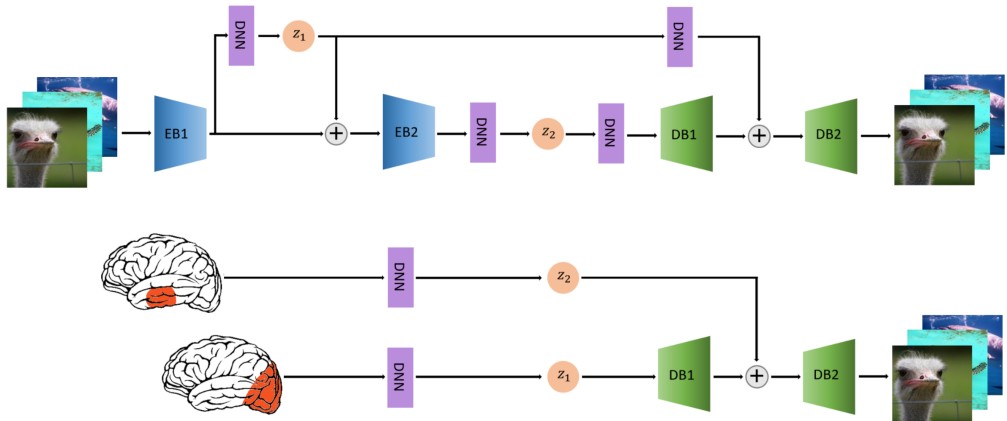

Figure 2: **Outline of our method**: a) We pretrain a Hierarchical Variational Autoencoder on a large set of images. Two layers of latent variables $\mathbf{z}_1, \mathbf{z}_2$ are inserted after each encoder (EB) and decoder (DB) block. b) We train the Neural Decoder by discarding the encoder from the previous step and learning a map from the fMRI voxels to the hierarchical latent space. The lower visual cortex (V1, V2, V3) is mapped to $\mathbf{z}_1$ and the higher visual cortex (FFA, PPA, LOC) to $\mathbf{z}_2$.

*Lower Bound* (*ELBO*), on the log-likelihood, as follows

$$\log p_\theta(\mathbf{x}) \geq \mathbb{E}_{q_\phi(\mathbf{z}|\mathbf{x})}\Big[\log \frac{p_\theta(\mathbf{x}, \mathbf{z})}{q_\phi(\mathbf{z}|\mathbf{x})}\Big] = \mathcal{L}(\theta, \phi; \mathbf{x})$$
$$= -KL(q_\phi(\mathbf{z}|\mathbf{x}||p_\theta(\mathbf{z}))) + \mathbb{E}_{q_\phi(\mathbf{z}|\mathbf{x})}[\log p_\theta(\mathbf{x}|\mathbf{z})], \tag{1}$$

where KL is the Kullback-Leibler divergence. The encoder and decoder are implemented by deep neural networks and their parameters are jointly optimized using gradient descent on the ELBO criterion. Similarly to standard VAEs, the reparametrization trick (26; 27) is used to allow us to back-propagate the gradient thought the stochastic sampling involved in the computation of Eq. 1.

## 3.2 Neural Decoder

We now leverage the latent space of the HVAE to learn a set of maps from the fMRI voxel space to the hierarchical latent variables. In more detail, each *region of interest* (*ROI*) is mapped via a dense neural network to a specific subset of the latent space. Brain regions in the earlier states of the visual pathway are mapped to the earlier layers of the latent hierarchy, whereas voxels from the higher visual cortex areas are mapped to the deep layers in the latent hierarchy. We assume that the HVAE has $L$ groups of latent variables $\mathbf{z}_1, \mathbf{z}_2, \ldots \mathbf{z}_L$ and that the fMRI voxels are partitioned into $n$ non-overlapping brain regions of choice, i.e., $\mathbf{y}_1, \mathbf{y}_2, \ldots \mathbf{z}_L$. Formally, the *Neural Decoder* is a set of maps from the $i$-th brain region to the $i$-th group of latent variables. Each of these maps is represented by a deep neural network with parameters $w_i$, i.e., $\mathbf{z}_i = \psi_{w_i}(\mathbf{y}_i)$, $i = 1, 2, \ldots L$. The reconstruction $\hat{\mathbf{x}}$ is then obtained by passing the latent variables $\mathbf{z} = \{\psi_{w_1}(\mathbf{y}_1), \psi_{w_2}(\mathbf{y}_2), \ldots \psi_{w_L}(\mathbf{y}_L)\}$ through the decoder model $p(\mathbf{x}|\mathbf{z})$ defined in Sec. 3.1.

The loss function used for training the Neural Decoder is an important design choice. Classic per-pixel measures, such as Euclidean distance, commonly used for regression problems, or the related Peak Signal-to-Noise Ratio (PSNR), are insufficient for images, as they assume pixel-wise independence. Therefore, to encourage the Neural Decoder to learn reconstructions guided by human visual perception, we use a *perceptual loss*. Perceptual loss is a class of loss functions that relies on the fact that CNNs extract hierarchical features. More specifically, deep features trained on supervised, self-supervised and unsupervised objectives are an effective model of human visual perceptual similarity (11). For a given image $\mathbf{x}$ and its reconstruction $\hat{\mathbf{x}}$, their perceptual loss is:

$$l(\mathbf{x}, \hat{\mathbf{x}}) = \sum_l \frac{1}{H_l W_l} \sum_{h,w} ||b_l \odot (f_{\mathbf{x}}^l - f_{\hat{\mathbf{x}}}^l)||_2^2, \tag{2}$$

where $f_{\mathbf{x}}^l, f_{\hat{\mathbf{x}}}^l$ are the layer-wise activations of a given, pretrained CNN model, $b_l \in \mathbb{R}^{C_l}$ is a channel-wise scaling vector. Intuitively, the perceptual loss in Eq. 2 extracts features for both the target and reconstructed image and then compares the features layer-wise using the Euclidean norm. To ensure that no bias is introduced during learning, it is important that the CNN used for evaluating Eq. 2 is different than the one used for the encoder. In our implementation we use a pretrained AlexNet as well as the code provided by Zhang et al. (11) to compute the perceptual loss.

## 3.3 MODEL TRAINING

For the **encoder** part of our HVAE, we use a pretrained VGG-19 model (28). This is a deep convolutional neural network of 19 convolutional layers and 3 fully connected layers. We use the weights from the model pretrained on ImageNet and discard the fully connected layers. We introduce latent variables by taking the output of a given convolutional layer, flattening it, passing it through a fully connected layer and, finally, through a variational layer which outputs the latent variable. This latent variable is re-sampled to avoid any dimension mismatch, and rerouted back to the main block, where it is aggregated with the output of the convolutional layer. Depending on how many latent layers we would like to insert, their exact position may vary. As an empirical design choice we choose to insert the latent layers equally spaced and after a convolutional block. A latent layer is always inserted at the output of the penultimate convolutional block.

The **decoder** part of our HVAE transforms the hierarchical latent variables to output images and consists of 4 transposed convolutional layers. The number of decoder filters are $[128, 64, 32, 16, 3]$ and all kernel sizes are set to $5$. Each transposed convolutional layer is followed by a 2d batch normalization and a ReLU non-linearity. The output of each transposed convolutional layer is interleaved with the latent variables. More specifically, each latent variable is initially passed thought a fully connected layer, re-sampled to avoid dimension mismatch and then aggregated with the output of the corresponding transposed convolution. Similarly to the encoder, we insert the latent variable such that we ensure symmetry and we always insert the penultimate latent variable before the first transposed convolution.

We start the training process by first deciding the number and position of the latent layers. The choice is guided by the type of fMRI data that we have as well as the level of latent space coarse-graining that we can achieve. For instance, if our fMRI data contains only the primary (V1) and the secondary (V2) visual cortex then we have two choices: a) we can either consolidate all voxels into a single vector and have a single latent layer in our HVAE or b) we can have two vectors containing the voxels from each brain area and train the HVAE such that it has two latent layers $\mathbf{z}_1, \mathbf{z}_2$ (example shown in Fig. 2). Naturally, if our fMRI data are more fine grain, we can add additional latent layers.

Following this design choice, the training proceeds in two phases: In the first phase, we pretrain the HVAE via self-supervision using the ELBO loss function Eq. 1 on a large ensemble of 50,000 real-world images from the ImageNet database. These images come from the same categories as the images shown to the subjects but no test images are included. This phase gives us meaningful latent representations and allows the HVAE decoder to adapt to the statistics of a large set of real-world images. In the second phase, the HVAE encoder is discarded, the HVAE decoder is kept fixed and the Neural Decoder is trained on supervised {fMRI, Image} pairs using the perceptual loss function Eq. 2. In this phase, we essentially learn a map from the voxels of each brain area to the corresponding latent layer and then use that latent vector to reconstruct the image.

## 4 EXPERIMENTAL RESULTS

To evaluate the utility of our method in practice, we carry out a series of experimental simulations. To measure the performance of our method, we use both qualitative comparisons of the reconstructions as well as quantitative metrics. In what follows, we give the details of the dataset used, the metrics implemented and baseline comparisons.

**Dataset**: We applied our pipeline on a commonly used, publicly available dataset known as **Generic Object Decoding** (**GOD**). The dataset consists of high-resolution ($500 \times 500$) stimuli images and their corresponding fMRI recordings. There exist 1250 (1200 train, 50 test) stimuli images selected from 200 object categories from the ImageNet database and the fMRI recording were obtained while 5 healthy subjects were viewing the stimuli (presentation experiment). The train- and test-fMRI data

consist of 1 and 35 (repeated recordings) per presented stimulus image, respectively. We use the post-processed fMRI data provided by Horikawa et al. (8), which contain voxels from 7 brain areas (V1,V2,V3,V4,FFA,PPA,LOC). The temporal component of the fMRI signal is averaged-out and the input to the model is a high-dimensional voxel vector. Even though there may be more comprehensive datasets, such as the BOLD 5000 (29) and the NSD (30) datasets (which in fact contain a higher number of more diverse images), we choose to focus on GOD for two primary reasons: 1) the dataset provides post-processed fMRI data, and 2) it has been used in numerous past studies (9; 1; 13; 14). Both of these facts facilitate the easy and fair comparison between different methods.

**Ablation Study**: We perform an ablation study, with the number of hierarchical layers and, consecutively, the number of brain regions, being the ablated parameter. Motivated by the two stream hypothesis (Sec 2) for the neural processing of visual information , we consider the following variants:

1. **Naive Baseline** (**NB**): We consider only one latent layer $\mathbf{z}_{NB}$ and all fMRI voxels are mapped to $\mathbf{z}_{NB}$. There are approximately 5000 voxels in this variant.

2. **Primary-Secondary** (**PS**): We consider 2 latent layers $\mathbf{z}_{V1}, \mathbf{z}_{V2}$ and the voxels from $V1, V2$ are mapped to the corresponding latent layer. There are approximately 1500 voxels.

3. **Dorsal Pathway** (**DP**): We consider the 3 latent layers $\mathbf{z}_{V1}, \mathbf{z}_{V2}, \mathbf{z}_{V3}$ and voxels from $V1, V2, V3$ are mapped to the corresponding latent. There are approximately 2500 voxels.

4. **Ventral Pathway** (**VP**): We consider 4 latent layers $\mathbf{z}_{V1}, \mathbf{z}_{V2}, \mathbf{z}_{V4}, \mathbf{z}_{PF}$ and the voxels from $V1, V2, V4, \{FFA, PPA\}$ are mapped to the corresponding latent layer. The voxels from $FFA$ and $PPA$ are merged to a single area. There are approximately 3300 voxels.

We note that by using different ROIs and/or by combining them to form different latent architectures, it is possible to obtain different ablated variants. We empirically noticed that by including the LOC, either concatenated as part of the latest latent layer of the VP or by creating a new LOC-only latent layer, there was no further performance improvements, only losses in terms of computational cost. Therefore, we restrict our exposition to the aforementioned 4 variants.

**Metrics**: The reconstruction quality is assessed both subjectively, i.e., by visual inspection of the output test images and comparison with the ground truth, as well as objectively. Our quantitative evaluation relies on metrics that encode the spatial dependence such as the Pearson Correlation Coefficient (PCC) and the Structural Similarity Index Measure (SSIM).

**Pearson Correlation Coefficient** (**PCC**): This metric is extensively used in statistics to measure the linear dependence between variables. In the context of image similarity, PCC is computed on the flattened representations of the two images. The limitation of PCC is its sensitivity to edge intensity or misalignment, which makes the metric assign larger value to blurry images (9).

**Structural Similarity Index Measure** (**SSIM**): Wang et al. proposed SSIM in (31) as a metric that quantifies the characteristics of human vision. Given a pair of images $p, q$, SSIM is computed as a weighted combination of *luminance*, *contrast* and *structure*. Assuming equal contribution of each measure, SSIM is first computed locally in a common window of size $N \times N$, and then the global SSIM is computed by averaging the SSIM over all non-overlapping windows.

These image similarity metrics defined are used for computing the *correct identification rate* in an **n-way classification** task. Let $M \in \{PCC, SSIM\}$ be a metric of choice, $\hat{p}_i$ be a reconstructed image and $P_i$ be a set containing the ground truth $p_i$ and a set of $n-1$ randomly selected target images. The *Correct Identification Rate* (*CIR*) is defined as follows:

$$CIR^n_M = \frac{1}{N} \sum_{i=1}^{N} \mathbf{1}\big(i = \arg\max_{p_j \in P_i} M(\hat{p}_i, p_j)\big), \tag{3}$$

where $N$ is the total number of images and the indicator function $\mathbf{1}(\cdot)$ has the value of 1 if the argument is true and 0 otherwise. The $CIR^n_M$ metric is essentially the frequency at which a reconstructed image can correctly identify the ground truth among $n-1$ randomly selected additional images. The chance level is $1/n$.

**Main Results**: We compare the performance of our method against several state of the art methods (SOTA) for the problem of neural decoding. The competitor methods are: the encoder-decoder based self-supervised method by Belyi et al. (9), the end-to-end, GAN-based pipeline by Shen et al. (14),

the GAN-conditioned method by Shen et al. (1) and the shape-schematic GAN by Fang et al. Figure 3 shows qualitative results and compares our method against the aforementioned competitors. All displayed images were reconstructed from the test-fMRI dataset. To improve the signal-to-noise ratio, the test fMRI test samples are averaged across trials. The results shown were obtained using the Ventral Pathway variant, which gave the best performance. We directly use the reconstructions reported in the respective papers by the authors. Our method tends to consistently produce more faithful reconstructions. Note that, even though the GAN-based decoders tend to produce more natural images, the reconstructions may deviate significantly from the stimulus image. This is because the GAN is introduced as an imaged prior, as noted by Belyi et al. (9). On the contrary, our method reconstructs the stimuli more faithfully, albeit the reconstructions appearing as a noisier version of the ground truth. (13).

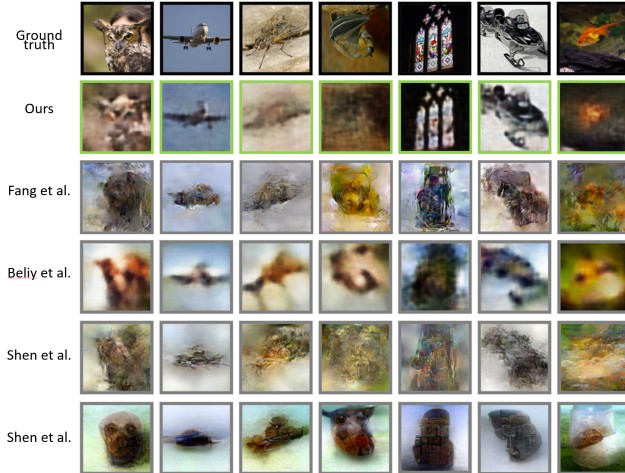

Figure 3: Qualitative comparison of reconstruction quality.

It is evident that the qualitative comparison highlights a trade-off between the naturalness of the reconstructed stimuli and the pixel-wise noise introduced in the reconstructions. To resolve the ambiguity, we perform an additional quantitative comparison using the $CIR^n$ metric. For this part we compare against the method by Belyi et al. (9) as well as the two variants of the method by Shen et al. (1). We directly compare against the results as reported by the authors of (9). The results are shown in Fig. 4. For our method, we report the correct identification rate obtained using the Ventral Pathway variant and we average across the metrics ($CIR^n_{PCC}$ and $CIR^n_{SSIM}$). We observe that our method consistently outperforms the competitors and, particularly in the 5-way and 10-way case, by a substantial margin. Additionally, we observe our method shows a small performance drop as we increase $n$, i.e, from 90% in the 2-way case to 79% in the 10-way case, whereas the performance loss for the competitor method is substantially higher. This performance is due to the following fact: Even though our method gives noisier reconstructions than the competitors, the high-level features such as color, texture and shapes are retained and, therefore, the task of identifying the correct ground truth from the reconstruction is easier. In contrast, please observe in Fig. 3 that the competitor methods may substantially alter the color or texture of the image, therefore leading to more frequent ground truth misidentification.

In the next experiment, we evaluate the decoding performance of different visual pathways. The results are shown on Fig. 5. Qualitatively, the ventral stream seems to be producing the best reconstructions, which is expected from a neuroscience perspective, given that this pathway's purpose is for visual perception and contains high level areas (FFA-PPA) for object recognition. Interestingly enough, even though the Naive Baseline contains all the available brain areas, the reconstruction quality is inferior, especially in the 2nd and 3rd images, which are far more complex.

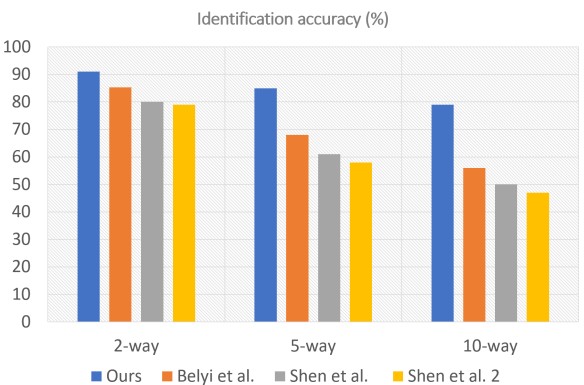

Figure 4: Correct identification ratio.

The V1-V2, Dorsal and Ventral variants essentially partition the brain areas into (progressively finer) segments and map the voxels from each area onto the hierarchical latent space of the HVAE decoder. Even though the increased performance among these variants may be partially explained by the

fact that the number of voxels increases, the main point of comparison should be against the Naive Baseline. The three models, PS, DP, and VP, are hierarchical, whereas the naive baseline includes all data but has no hierarchy. Simply the fMRI responses from two regions, V1 and V2 and discarding all other voxels we are able to achieve better performance than simply mapping all voxels in a big latent vector. This suggests that the hierarchy is far more important than the amount of data that we fed to the model. This is in line with previous studies which concluded that models trained on the whole visual cortex perform slightly worse than those trained on separate areas (13).

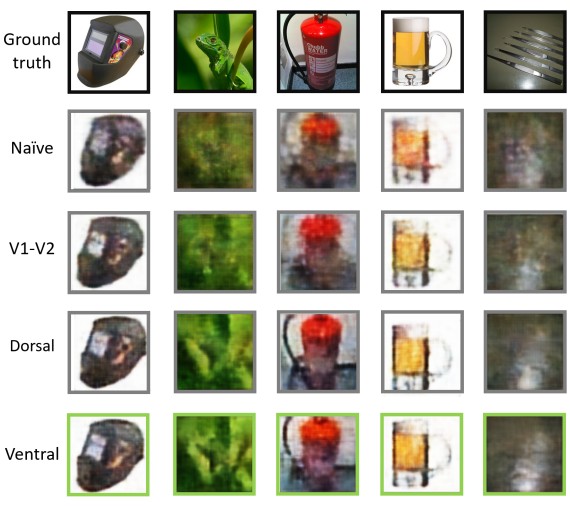

Figure 5: Qualitative comparison for different pathways.

Additionally, since the Naive Baseline essentially learns a map from *all voxels* to a *single* latent layer, it is natural to assume that is fails due to massively overfitting. However, if overfitting is indeed the only reason for that failure, we would expect the reconstruction performance to decrease as we add more voxels to the model input. However, the figure shows the exact opposite: the performance increases as we add more voxels. This suggests that overfitting is not the only reason for the Naive model's failure and that the the hierarchical structure of the visual information processing needs to be explicitly taken into account. However, one may hypothesize that the performance increase in the Ventral Pathway model may come from the partitioning of the ROIs and that the hierarchical structure has little impact. To test this, it is prudent to include a variant in which the VP ROIs are randomly shuffled to assess whether the hierarchical structure or the partitioning of the voxels drives the performance. We call this variant **VP Permutations** and it supplements the previous 4 variants.

| | $CIR_{PCC}^2$ | $CIR_{SSIM}^2$ | $CIR_{PCC}^5$ | $CIR_{SSIM}^5$ | $CIR_{PCC}^{10}$ | $CIR_{SSIM}^{10}$ |
|---|---|---|---|---|---|---|
| Naive Baseline | 0.77 | 0.78 | 0.64 | 0.66 | 0.57 | 0.58 |
| Primary-Secondary | 0.80 | 0.82 | 0.72 | 0.73 | 0.65 | 0.67 |
| Dorsal Pathway | 0.88 | 0.90 | 0.81 | 0.80 | 0.75 | 0.75 |
| Ventral Pathway | **0.91** | **0.92** | **0.84** | **0.85** | **0.79** | **0.79** |
| VP Permutations | 0.79 | 0.80 | 0.65 | 0.66 | 0.60 | 0.58 |

Table 1: The $n$-way correct identification rate ($n = 2, 5, 10$) for all ablated variants using the Pearson Correlation Coefficient (PCC) and the Structural Similarity Index Measure (SSIM) as a selection criterion. We report the mean across subjects. The results for VP Permutations are averaged across 4 permutations. The inter-subject deviation was in the range of $0.02 - 0.05$. The chance levels are $0.5, 0.25, 0.10$, respectively.

Following that, we present quantitative results on Table 1. On this table we give the the $n$-way correct identification rate $CIR^n$ for $n = 2, 5, 10$, for all ablated variants and the VP Permutations for both metrics (PCC and SSIM). The results on this table validate the aforementioned qualitative observations. The identification accuracy is progressively increasing as we partition the brain into more fine areas and as we add hierarchical layers in the HVAE onto which the brain areas are mapped. Additionally, we observe that the newly introduced VP Permutations variant leads to performance degradation, which suggests that the hierarchy and not the partitioning drives the performance result. However, we do note that we have a slight performance increase compared to the Naive Baseline, which indicates that merely partitioning the brain regions is beneficial, albeit not as beneficial as accounting for the hierarchy.

To verify that our method can be successfully applied to all subjects and study potential inter-subject variation of the results, we show in Fig. 6 the learning curves for all 5 subjects and for all metrics $CIR_M^n$, $n = 2, 5, 10$ and $M \in \{PCC, SSIM\}$. The metrics were calculated using the test samples

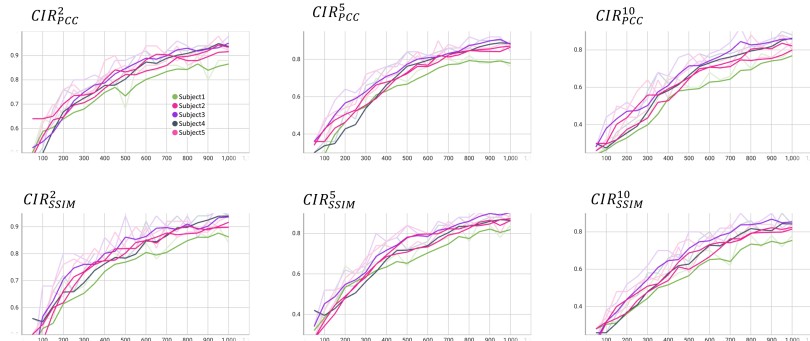

Figure 6: Learning curves for $CIR_M^n$, $n = 2, 5, 10$ and $M \in \{PCC, SSIM\}$ across all subjects. The horizontal axis is the number of epochs. Subject 3 is marginally outperforming the other subjects and Subject 1 gives the worst performance (*figure best viewed in color*).

and the ventral pathway variant. Even though the metrics appear similar across subject, after careful examination of the curves some subtle discrepancies and trends can be observed. Subject 1 is consistently performing approximately 5% worse across all metrics whereas Subject 3 is marginally outperforming the other subjects by 2%. The fact that the Subject 3 gives the best reconstructions has been verified in previous studies (10) and is attributed to differences in the signal-to-noise ratio across subjects. Finally, Fig. 6 allows us to study how training progresses and validate that no overfitting occurs. We observe that, in all cases, the metrics saturate at about 800 epochs, which gives us an empirical estimate of how many iterations our model needs to achieve good performance.

Finally, we present on Fig. 7 a qualitative comparison against the recent method by Lin at al. (21). Even though the images are different, we can extract some interesting conclusions. Our method seems to be more faithful albeit having lower reconstruction quality. The method by Lin et al. has excellent reconstruction quality and only losing some elements of the faithfulness, which are not hurting the schematic understanding. This may be because the later method is more sophisticated and handles schematic information in addition to the visual as part of the modeling procedure. Since our method is not incorporating any schematic information, an interesting future direction is to extend the model in order to see if our pipeline can achieve similar quality.

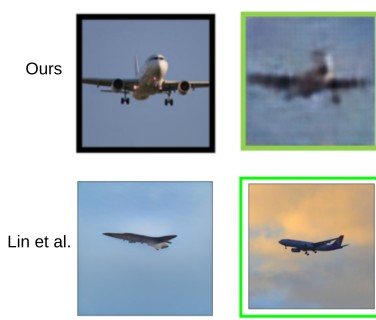

Figure 7: Qualitative comparison with Lin et al. (21) on schematically-similar test images.

## 5 CONCLUSION

We addressed the problem of neural decoding from fMRI recordings and proposed a novel architecture inspired by neuroscience. More specifically, motivated by the fact that the human brain processes visual stimuli in a hierarchical fashion, we postulated that this structure can be captured by latent space of a hierarchical variational autoencoder (HVAE). Our HVAE serves as a proxy to learning meaningful latent representations of stimuli images and can be pretrained on a large dataset of high-resolution images. Following that, we train our Neural Decoder to learn a map from the fMRI voxel space to the HVAE latent space. Our architecture replicates the visual information processing in the human brain in the sense that earlier visual cortex areas (e.g., primary-secondary visual cortex) are mapped to the earlier latent layers, whereas voxels from the higher visual cortex (e.g., PPA, FFA areas) are mapped to the later latent layers. We validated our approach using fMRI recordings from a visual presentation experiment involving 5 subjects and compared against other methods. Our work paves the way to constructing better models to replicate human perception and understanding the nuances of human visual reconstruction, both of which could utilized to better understand the brain, assist people with visual disabilities and perhaps in decoding imagery during sleep.

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
