# OpenReview forum: "Neural Decoding of Visual Imagery via Hierarchical Variational Autoencoders"
_ICLR.cc/2023/Conference — Submitted to ICLR 2023_

### Official Review · Reviewer_sMBW · 2022-10-23

**Confidence:** 4
**Correctness:** 2
**Technical Novelty And Significance:** 2
**Empirical Novelty And Significance:** 2
**Recommendation:** 3

**Clarity, Quality, Novelty And Reproducibility:**

The novelty this paper is modest but easy to understand. However, major flaw with respect to data using exists which leads to unbelievable experimental results.  The paper contains a large number of informal expressions and clerical errors, the structure is not clear, and the layout is confusing. According to this, the clarity and quality of the paper are  poor. The reproducibility is poor since the code is not available.

**Strength And Weaknesses:**

Strength:
1. This is the first approach that uses HVAEs in the context of neural decoding.
2. The proposed method combines decoding models with neuroscience priors and the motivation makes sense.
3. The experiments are relatively sufficient.


Weaknesses:
(major)
1. Experiments have serious data leakage problem. The HVAE were pretrained in ImageNet and the dataset GOD you used is still collected based on ImageNet. Accordingly, I think the results of this paper is unbelievable.
2. Although authors has pointed out why they only used GOD dataset, there is still a lot of preprocessed datasets that can be used such as BOLD 5000 and NSD. NSD dataset is collected based on MS-COCO which can avoid aforementioned problem.
3. The compared methods are not comprehensive. I suggest that the authors add the method proposed in "Mind Reader:
Reconstructing complex images from brain activities"  in which the reconstruction quality is relatively better than the compared methods you have chosen.
4. The article introduced less content about HVAE, formula (1) is even traditional VAE, and used a lot of space introducing the background and evaluation indicators of neuroscience.  By the way, formula (1) and formula (2) needn't two labels.


(minor)
1. The writing of this paper is terrible.  A major refine of this manuscript is necessary.  Informal expressions such as "{fMRI,Image} pairs" and "Naturally, if our fMRI data are more fine grain, we can add additional latent layers" should be avoided.  Typos such as text in line 3 of page 5 should be corrected. In the experimental results, text wrapping charts is not acceptable for readers.  The structure of the article needs a careful adjustment such as the last two paragraphs of 3.1 should be placed into experimental details.
2. The main results part should contain a series of subtitle for easy understanding.

**Summary Of The Paper:**

The authors developed a natural images reconstruction model from fMRI with hierarchical variational autoencoders. Their model achieved better reconstructions compared to the state of the art and our ablation study indicates that the hierarchical structure of the latent space is responsible for that performance.
The main contributions are as follows: 1. The paper leveraged the hierarchical manner of human brain during visual stimuli and proposed a novel architecture with hierarchical variational autoencoders to reconstruct neural images. 2. Their architecture replicates the natural hierarchy of visual information processing in the latent space of a variational model. 3. Their experimental analysis suggests that hierarchical latent models provide better priors for decoding fMRI signals.

**Summary Of The Review:**

The work applies an existing method to an existing task. The novelty this paper is modest but easy to understand. My major concern is the flaw with respect to data using exists which leads to unbelievable experimental results. Besides, the writing of this paper is terrible. I think the quality of the paper is not up to the average quality of the papers accepted by ICLR. I suggest the authors should change their experimental data and carefully revise the manuscript.

---

> ### Author Response · Authors · 2022-11-14
> **Response to Reviewer**
>
> We would like to thank the review for reading our paper and for giving feedback. Please find our responses below.
>
> >Experiments have serious data leakage problem. The HVAE were pretrained in ImageNet and the dataset GOD you used is still collected based on ImageNet. Accordingly, I think the results of this paper is unbelievable.
>
> Indeed, pretraining the HVAE in the entire ImageNet would cause data leakage as the GOD images come from ImageNet dataset as well. Please note that we did not train our HVAE in the entire ImageNet, it would be unnecessarily costly to do so. For the encoder part of our HVAE, we used pre-trained weights from the AlexNet architecture. Then, we pretrained our HVAE using 50.000 ImageNet images (not including the test image). Finally, we discard the HVAE encoder and train our neural decoder on the GOD train set. The HVAE never sees an image from the GOD test dataset. Please note that this training procedure is very similar and inspired by the work of Beliy et al, 2019. We aggregate all the training details in the 3.3 subsection. Admittedly, these details were slightly spread out in the paper.
>
> > Although authors has pointed out why they only used GOD dataset, there is still a lot of preprocessed datasets that can be used such as BOLD 5000 and NSD. NSD dataset is collected based on MS-COCO which can avoid aforementioned problem.
>
> Thank you for noting that. These dataset are indeed very good. Unfortunately, we do not have the capacity to re-run our experiments on these datasets and restructure the paper to include the new results within the rebuttal period. However, we have included references to the datasets in the paper to emphasize that GOD is not the only option.
>
> > The compared methods are not comprehensive. I suggest that the authors add the method proposed in "Mind Reader: Reconstructing complex images from brain activities" in which the reconstruction quality is relatively better than the compared methods you have chosen.
>
> Thank you for sharing that paper. This is a very interesting paper with impressive results. We have referenced the paper in Sec.2  and added a comparison in page 9. (Note: The comparison on page 9 was supposed to be highlighted for your ease of reference but the \hl was giving us trouble on that part. )
>
> > The article introduced less content about HVAE, formula (1) is even traditional VAE, and used a lot of space introducing the background and evaluation indicators of neuroscience. By the way, formula (1) and formula (2) needn't two labels.
>
> Thank you, we removed the two labels from the equation.
>
> > The writing of this paper is terrible. A major refine of this manuscript is necessary. Informal expressions such as "{fMRI,Image} pairs" and "Naturally, if our fMRI data are more fine grain, we can add additional latent layers" should be avoided. Typos such as text in line 3 of page 5 should be corrected. In the experimental results, text wrapping charts is not acceptable for readers. The structure of the article needs a careful adjustment such as the last two paragraphs of 3.1 should be placed into experimental details.
>
> We are really sorry that you found our paper hard to read. We fixed the things that you (and the other reviewers) suggested. We hope that the updated version of the paper is more clear to you. If not, we would be happy to make any further modifications that you deem necessary to improve the clarity..
>
> > The main results part should contain a series of subtitle for easy understanding.
>
> Could you please elaborate on where the subtitles should be and what they should contain? We’ll be happy to include them!

---

> > ### Comment · Reviewer_sMBW · 2022-11-30
> > **Review after reading the author response**
> >
> > Based on feedback from the authors and comments from other reviewers, I keep my rating unchanged. The technical contribution of this paper is small. Although the image reconstruction is clear, it is mainly due to the role of HVAE, and the author has not made new contribution to fMRI signal processing. Overall, I don't think this paper is valuable enough for either the machine learning community or the neuroscience community.

---

> ### Author Response · Authors · 2022-11-24
> **Clarification on code availability**
>
> > The reproducibility is poor since the code is not available.
>
> Please note that the code of the paper is available under the supplementary material. It has been available since the initial paper submission.

---

### Official Review · Reviewer_eDwx · 2022-10-24

**Confidence:** 4
**Clarity, Quality, Novelty And Reproducibility:** Clear enough and well written.
**Correctness:** 3
**Technical Novelty And Significance:** 3
**Empirical Novelty And Significance:** 4
**Recommendation:** 6

**Strength And Weaknesses:**

Strength: Their image decoding results are very impressive, far better than state-of-the-art.
Weakness: The insights are perhaps obvious, and the technical innovation is relatively minor.

The comparison between V1/V2 versus ventral and dorsal is not completely fair and problematic.  These comparisons are confounded with different numbers of parameters, and thus the conclusion about the contribution of the different visual areas, ventral and dorsal streams are  potentially problematic.  It seemed that the more neural responses are used, the best would be the performance. For instance, if they include V3 in the ventral stream's model, would  the performance be even better?   To their credit,  they did show that the hierarchy is important, and their argument that the critical comparison is within the naive baseline which has the same number of parameters is not unreasonable.

**Summary Of The Paper:**

The authors used a hierarchical VAE to learn representations from natural images and leverage their latent space hierarchy to learn voxel-to-image mappings. They showed that mapping V1/V2 responses to the early layer and higher visual areas to the deep layer of the latent hierarchy allow them to achieve superior image decoding performance than state-of-the-art approaches.

**Summary Of The Review:**

This paper achieved the best performance in image decoding based on fMRI signals. The motivation is sound, the idea obvious. The technical innovation is minor.  it would become the new baseline for future algorithms to compare against. The comparison between V1/V2, ventral and dorsal streams are potentially problematic and incomplete.

---

> ### Author Response · Authors · 2022-11-14
> **Response to Reviewer**
>
> We would like to thank the reviewer for reading our paper and giving us feedback. Please find our responses to your comments bellow:
>
> > Strength: Their image decoding results are very impressive, far better than state-of-the-art. Weakness: The insights are perhaps obvious, and the technical innovation is relatively minor.
>
> Thank you for acknowledging the significance of our results. Indeed, the technical innovation is not outstanding,
>
> > The comparison between V1/V2 versus ventral and dorsal is not completely fair and problematic. These comparisons are confounded with different numbers of parameters, and thus the conclusion about the contribution of the different visual areas, ventral and dorsal streams are potentially problematic. It seemed that the more neural responses are used, the best would be the performance. For instance, if they include V3 in the ventral stream's model, would the performance be even better? To their credit, they did show that the hierarchy is important, and their argument that the critical comparison is within the naive baseline which has the same number of parameters is not unreasonable.
>
> This is indeed correct. Directly comparing V1-V2 against the ventral or dorsal pathway is not fair due to having different number of parameters. Please note that we do not aim to directly compare the aforementioned but rather to highlight the importance of hierarchy: By simply breaking down to just V1/V2 as discarding all other areas we are able to obtain slightly better performance than simply mapping all brain areas to a single latent vector. We have highlighted that fact at the end of page 7.

---

### Official Review · Reviewer_J68Z · 2022-10-24

**Confidence:** 5
**Correctness:** 2
**Technical Novelty And Significance:** 2
**Empirical Novelty And Significance:** 2
**Recommendation:** 1

**Clarity, Quality, Novelty And Reproducibility:**

As noted above, there are crtitical questions about the methodology that are not clearly described.

fMRI data have very low quality and signal-to-noise ratio.


**Strength And Weaknesses:**

Strengths

It is interesting to link neural representations to machine learning representations, if done correctly.

Weaknesses
The title is somewhat misleading. While the word “imagery” can be used to denote the processing of images, a lot of people in the community have used the word to denote visual imagination, which is not what this study is about. The authors briefly mention decoding dreams and such in the conclusion as potential future work, but this is not what is shown here.

It is unclear what the goal of the algorithm should be. One goal could be to attempt to reconstruct images from fMRI voxel activity. The qualitative comparison of perhaps cherry-picked examples is not a convincing result and certainly not a rigorous way of presenting results. The authors present some quantitative results in Fig. 4, but they do not define what 2-way, 5-way, and 10-way represent. More critically, why is it relevant to reconstruct images? The purpose of the visual system is to extract useful information, not to reconstruct images. One should get a much better reconstruction from the activity of neurons in the retina. Whatever cortex does, it is not reconstruction.

A central issue is to understand what goes into the training set and what goes into the test set. The first question is whether any of the results represent cross-validation across images. My understanding is that the authors are only using cross-validation across trials, which is not very interesting. Any powerful enough model can memorize the dataset. In this case, the only question is how consistent the data are across trials. If this is the main point, it would be useful to show the degree of consistency in the data across trials, showing the actual data across repetitions and measures of self-consistency.

A different question, and a more interesting one is to perform cross-validation across very different images. In this case, there are a lot of questions that should be addressed. First, are the images in the training set and test set from the same categories (i.e., same imagenet labels)? How different are the images in the training set and the test set? How well can simple classifiers match the images in the test set to those in the training set? These are critical questions to assess what the models are doing or trying to do.



**Summary Of The Paper:**

This study analyzes existing functional magnetic resonance imaging (fMRI) data from human participants that passively viewed a series of images. The analysis is based on using hierarchical variational autoencoders and then using fitting to map the activations of units in the model to the activation of voxels in order to reconstruct the images. Compared to a few baselines, the authors claim to obtain better qualitative results in reconstructions and better quantitative results in discriminating images across trials without cross-validation across images.

**Summary Of The Review:**

This is an effort to try to reconstruct images from fMRI data. Although the proposed method performs better than the baselines suggested by the author, it is unclear whether this can really extrapolate and provide any real insights about the underlying representations

---

> ### Author Response · Authors · 2022-11-14
> **Response to Reviewer**
>
> > A central issue is to understand what goes into the training set and what goes into the test set. The first question is whether any of the results represent cross-validation across images. My understanding is that the authors are only using cross-validation across trials, which is not very interesting. Any powerful enough model can memorize the dataset. In this case, the only question is how consistent the data are across trials. If this is the main point, it would be useful to show the degree of consistency in the data across trials, showing the actual data across repetitions and measures of self-consistency.
>
> We actually did not do any cross-validation as the model does not have a complex set of hyperparameters to tune and we have a very limited amount of data to waste for hyperparameter tuning. We set basic hyperparemeters (batch size, learning rate, etc) to commonly used values and manually tuned them using the test set. The results reported on the paper come from the test set.
>
> We wanted to make sure that we did not make any misl eading statement regarding the training process so we scanned the paper and we could not find the word “cross-validation” mentioned anywhere. Perhaps the reviewer is confused with the aforementioned n-way classification task, as it is common to report “n-way cross-validation” results. Please read the end of page 6 to see the difference between them.
>
> What goes in the training/test set is mentioned in the “Dataset” paragraph in the experimental section. The dataset we use (GOD) is a well-known dataset in the field of neural decoding and the train/test images are already pre-defined.
>
> > A different question, and a more interesting one is to perform cross-validation across very different images. In this case, there are a lot of questions that should be addressed. First, are the images in the training set and test set from the same categories (i.e., same imagenet labels)? How different are the images in the training set and the test set? How well can simple classifiers match the images in the test set to those in the training set? These are critical questions to assess what the models are doing or trying to do.
>
> These are all very interesting questions. Unfortunately, we do not have the capacity and the time to investigate them in the context of this work as this would deviate from the goals of this paper.

---

> > ### Comment · Reviewer_H6z7 · 2022-11-14
> > **You used the test set to set hyperparameters??**
> >
> > "manually tuned them using the test set. The results reported on the paper come from the test set."
> > This is a big no-no. Did you mean this?

---

> > > ### Author Response · Authors · 2022-11-14
> > > **Response**
> > >
> > > If you are concerned about overfitting to the hyperpameters, please note that there is not that many hyperparameters to overfit. Only learning rate, batch size and number of epochs. The architecture of the model is fixed. We did not do any extensive tunning rather we just set them to commonnly used values (e.g., the batch size was in the range 64-128, and the learning rate of the optimizer at 0.0001) and tweaked them around those values to get the best performance. We tried very few hyperparameters and there was litle variation in the test set performance.  This was sort of necessary as we have very limited amount of data. And please note that the model never sees the test images during training.

---

> ### Author Response · Authors · 2022-11-14
> **Response to Reviewers**
>
> We would like to thank the reviewer for reading our paper and for the feedback. Please find our responses below:
>
> > Weaknesses The title is somewhat misleading. While the word “imagery” can be used to denote the processing of images, a lot of people in the community have used the word to denote visual imagination, which is not what this study is about. The authors briefly mention decoding dreams and such in the conclusion as potential future work, but this is not what is shown here.
>
> Thank you for that comment. Indeed, imagery may imply dreams. If it is allowed to, we will remove the word "imagery" from the title. It is not clear to us right now if it's allowed to change the tittle and we see no such option on OpenReview.
>
> > It is unclear what the goal of the algorithm should be. One goal could be to attempt to reconstruct images from fMRI voxel activity. The qualitative comparison of perhaps cherry-picked examples is not a convincing result and certainly not a rigorous way of presenting results. The authors present some quantitative results in Fig. 4, but they do not define what 2-way, 5-way, and 10-way represent. More critically, why is it relevant to reconstruct images? The purpose of the visual system is to extract useful information, not to reconstruct images. One should get a much better reconstruction from the activity of neurons in the retina. Whatever cortex does, it is not reconstruction.
>
> We understand your concern that presenting cherry-picked examples is not rigorous. Please note that the examples presented in the discussion surrounding Fig. 3 and 4 are not cherry picked. We present the exact same images as in the papers of Shen et al, Beliy et al, and Fang et al, which are well-known, state of the art contributions in the same problem.
>
> The 2-way, 5-way and 10-way, or more generally the n-way classification, is a classification task. Intuitively, we compute the distance between the model output and n test images (including the actual target) and we pick the one with the lowest distance. The idea is that we try to measure how often the model can identify the correct image in a population of n-1 randomly sampled additional images. This is described at the end of page 6. This metric is highly relevant as it gives us a proxy for assessing the reconstruction performance and for comparing different methods. It has been used by the aforementioned state of the art methods and, as previously, the results we report in Fig.4 are compared against the results reported by other papers using the same metric.
>
> Finally, we understand that the purpose of the visual system is to extract useful information and that the brain does not do image reconstruction internally. The goal of the paper is to reconstruct images from fMRI activity. This may not be aligned with the purpose of the visual system, however, it is a very well known problem with practical interest and an excessive amount of research work has been done to solve it. Please read the first paragraph of our paper and the references therein for an introduction in the problem of neural decoding.

---

### Official Review · Reviewer_H6z7 · 2022-10-26

**Confidence:** 4
**Correctness:** 3
**Technical Novelty And Significance:** 4
**Empirical Novelty And Significance:** 4
**Recommendation:** 8

**Clarity, Quality, Novelty And Reproducibility:**

In the abstract, you say: "The current architectures are bottlenecked because they fail to effectively capture the hierarchical processing of visual stimuli that takes place in the human brain. Motivated by that fact,..."
This is not a fact, it is an opinion. Please fix.

Figure 2b has a bug - you show the lower visual areas feeding into Z_2 and the higher ones into Z_1, which is the opposite of what the Figure caption says. Also, in the caption: "using by discarding" -> "by discarding." Also, there is no (a) or (b) in the figure. You could say "top" and "bottom" instead.

abstract, main text: You mention that you decode "natural" images and train the VAE on "natural" images. These images are not natural - that wording refers to images of natural scenes, like forests, woods, etc. Just remove the word "natural" everywhere.

Typos, wording:
1st line of page 3:  object -> objects
Fig 2 caption: using by -> by
sec 3.3, line 3: contain -> contains
page 6, line 3: thought -> though
line 7: fact -> facts
6th line from the bottom: "compares our method" - which method? I assume the ventral pathway. Again - why didn't you include LOC?
4th line from the bottom: you say you average the fMRI responses corresponding to the same category. I assume you mean corresponding to the same image, not category. If you are averaging responses over categories, that definitely needs more explanation.

Page 7, middle: "Qualitatively..." Qualitatively, the ventral and the dorsal both look about the same to me, and this is born out in the quantitative scores, which are very close.
5th line from the bottom: there -> three. But I would just remove the whole "i.e., we aim to compare..." phrase - it isn't needed. Then replace the sentence that starts "The former ones..." with: "The three models, PS, DP, and VP, are hierarchical, whereas the naive baseline includes..."
3rd line from the bottom: "by breaking down brain to two..." -> by breaking down the fMRI responses from two regions, V1 and V2, and discarding all other voxels, we are able to achieve better performance compared to mapping all the voxels into one latent vector.

Page 8: "Furthermore, we observe that the SSIM gives a modest but consistent performance gain compared to PCC." No! That just means that SSIM is a slightly more lax measurement. The two measurements are not directly comparable. There is no reason (based on the numbers) to prefer one to the other.
2nd line from the bottom: subject -> subjects.

**Strength And Weaknesses:**

This is an excellent paper, with excellent results.

Strengths:

+ The approach is well-motivated by reference to lower and higher brain areas corresponding to lower and higher layers of the decoder.

+ Although the above point has been made before, the novelty here is that the decoder is used to map to the latent variables at different levels of the hierarchical VAE, rather than to the feature layers of a CNN.

+ The results are very strong, both quantitatively and qualitatively.

+ The direct comparison to a factored model with the ROIs randomly assigned to the hierarchy is a nice demonstration that it is the hierarchy that matters, not the division into regions.

Weaknesses, with concrete, actionable feedback

- There are a few things that could be corrected easily, in terms of typos, etc. See below.

- why didn't you include LOC in the Ventral pathway model?

- The poorer results based on mapping all of the voxels to one latent might be a result of the LOC voxels being noisier. A fairer comparison would be to create three more baseline models, each one using the same areas as the three hierarchical models, but without the hierarchy. This would make a stronger argument for the approach.



**Summary Of The Paper:**

This paper presents a novel method for decoding images from FMRI, using a hierarchical VAE. First, a VAE is trained on images from imagenet. Then the encoder is removed, and a mapping is learned from the images to the latent variables of the VAE. The VAE is divided into lower and higher level areas (i.e., it's hierarchical), and lower and higher level brain recordings are used to feed into corresponding areas. The results are impressive, and to my knowledge, represent a new state of the art.

After reading the other reviews, and the authors' responses to them, I am going to reduce my score from 10 to 8, as they used the test set to set hyperparameters. While they claim that the hyperparameters were not particularly impactful on the results, this is a clearly big mistake in terms of ML practice and should be avoided at all costs. Nevertheless, I believe their results still hold (as they used pretty standard hyperparameters, apparently), and the hierarchical nature of their model clearly makes a difference, and is worth publishing.

After reading the response to my reduction in score, I'm not clear that the authors totally understand that hyperparameters should not be tuned on the test set. They seem to think it is ok to tune training hyperparameters using the test set. This is still not the case. Again, I think there is still much to like here.

**Summary Of The Review:**

I can find few weaknesses and many strengths in this work. There is novelty (mapping to latents rather than features), a nice comparison between using just V1 and V2, using the dorsal pathway, and using the ventral pathway, as compared to a non-hierarchical model that uses all of the voxels. However, the final paper should include comparisons with each of the three models using the same areas without the hierarchy.

Again, in summary, I am reducing my score based on the hyperparameter tuning.

---

> ### Author Response · Authors · 2022-11-14
> **Response to Reviewer**
>
> We would like to thank the reviewer for reading our paper and for his kind words. Please find our response to your questions below:
> > In the abstract, you say: "The current architectures are bottlenecked because they fail to effectively capture the hierarchical processing of visual stimuli that takes place in the human brain. Motivated by that fact,..." This is not a fact, it is an opinion. Please fix.
>
> Indeed this is a subjective, qualitative statement. We rephrased it to make it less strong: “However, current architectures fail to efficiently capture the hierarchical processing of visual information processing, which may bottleneck their representation capacity”
>
> > Figure 2b has a bug - you show the lower visual areas feeding into Z_2 and the higher ones into Z_1, which is the opposite of what the Figure caption says. Also, in the caption: "using by discarding" -> "by discarding." Also, there is no (a) or (b) in the figure. You could say "top" and "bottom" instead.
>
> We thank the reviewer for the suggestions and following his advice we fixed the issues on the figure.
>
> > abstract, main text: You mention that you decode "natural" images and train the VAE on "natural" images. These images are not natural - that wording refers to images of natural scenes, like forests, woods, etc. Just remove the word "natural" everywhere.
>
> We wanted to emphasize that the images are complex, real-world images. We removed the word “natural” and, in some places (especially in the introduction), we replaced it with “real-world” just to emphasize the aforementioned fact.
>
> > Typos, wording: 1st line of page 3: object -> objects Fig 2 caption: using by -> by sec 3.3, line 3: contain -> contains page 6, line 3: thought -> though line 7: fact -> facts 6th line from the bottom: "compares our method" - which method? I assume the ventral pathway. Again - why didn't you include LOC? 4th line from the bottom: you say you average the fMRI responses corresponding to the same category. I assume you mean corresponding to the same image, not category. If you are averaging responses over categories, that definitely needs more explanation.
>
> Thank you for pointing out the typos. We made the corrections. Regarding the averaging, we average across trials in the test set, not over categories (which in this case does not make any difference as there is only one image per category in the test set). We removed that statement to avoid confusion. Regarding the LOC, we actually did run some experiments with LOC included. We included it an part of the latest layer in VP variant as well as an additional latent layer. We observed no further improvement by including it. It seemed like the model has reached the capacity limit and further ROIs made no difference in performance, only hurt in the computational cost. We added a note in the “Ablation Study” section  where we discussed that.
>
> > Page 7, middle: "Qualitatively..." Qualitatively, the ventral and the dorsal both look about the same to me, and this is born out in the quantitative scores, which are very close. 5th line from the bottom: there -> three. But I would just remove the whole "i.e., we aim to compare..." phrase - it isn't needed. Then replace the sentence that starts "The former ones..." with: "The three models, PS, DP, and VP, are hierarchical, whereas the naive baseline includes..." 3rd line from the bottom: "by breaking down brain to two..." -> by breaking down the fMRI responses from two regions, V1 and V2, and discarding all other voxels, we are able to achieve better performance compared to mapping all the voxels into one latent vector.
>
> Thank you for these corrections, we put them in the paper.
>
> > Page 8: "Furthermore, we observe that the SSIM gives a modest but consistent performance gain compared to PCC." No! That just means that SSIM is a slightly more lax measurement. The two measurements are not directly comparable. There is no reason (based on the numbers) to prefer one to the other. 2nd line from the bottom: subject -> subjects.
>
> Correct! We thought about that statement again and you are right. We removed it was not really necessary.

---

> ### Author Response · Authors · 2022-11-23
> **Response to Reviewer**
>
> > After reading the other reviews, and the authors' responses to them, I am going to reduce my score from 10 to 8, as they used the test set to set hyperparameters. While they claim that the hyperparameters were not particularly impactful on the results, this is a clearly big mistake in terms of ML practice and should be avoided at all costs. Nevertheless, I believe their results still hold (as they used pretty standard hyperparameters, apparently), and the hierarchical nature of their model clearly makes a difference, and is worth publishing.
>
> It is indeed the case that tuning the hyperpameters using the test set may be a mistake. Our wording in our response to Reviewer J68Z was not correct.  Please read the following explanation for a detailed clarification.
>
> Whether or not it is acceptable to use the test set for hyperparameter tuning boils down to what types of hyperparameters you are tunning as well as how the tunning is embedded in the overall training process. Following the simple example [here](https://www.geeksforgeeks.org/svm-hyperparameter-tuning-using-gridsearchcv-ml/), suppose we are training an SVM clasifier using GridSearch for the hyperpameters. The correct way to do it is to train a different model for all combinations of hyperparameters and then use the validation set to pick the best hyperparameters. The wrong way would be to use the validation metric as a training signal. In the later case, we end up training on the validation set. There are two distinct characteristics in the above case:
>
> 1. The dimension of the hyper-parameter space is comparable to the dimension of the model parameter space. There is ~20 model parameters and 2 hyperparameters.
> 2. The hyperparameters are essentially model hyperparameters, i.e., they appear directly in the model definition. This means that different values can lead to vastly different train/validation performance.
>
> All of the above do not apply to our case for the following reasons:
>
> 1. We did not use any metric evaluated in the test set as a signal to guide the model training. What we did is that we picked some inital, reasonable values for the learning rate, batch size and number of epochs, trained on the train set and evaluated on the test set. Then, we varied those hyperparameters as part of our practical expiramentation in order to see how the model performs.
> 2. Our hyperparameter space is really small compared to the model parameter space. We only "tuned" 3 hyperparameters whereas our overall model has parameters in the range of 10^5-^10^6. Even if we were to selectivelly and intentionally overfit 3 parameters, one could hardly say that the model overfits given that a few parameters have little to no impact on the performance of such a complex model.
> 3. Lastly, and most importantly, the hyperparameters that we tunned were training hyperparameters, not model hyperparameters. This means that by varying our hyperparameters we expect (and observed) that the train/test performance will vary in a similar fashion. For example, by increasing the learning rate **both** the train and test loss converged faster but risked overshooting in some runs. By increasing the batch size, the time per epoch was lower but we required higher number of epochs to converge. The purpose of us tunning these hyperparameters was to study those trade offs and **not** to cherry-pick the hyperparameters that will give us the best results. You cannot really overfit to training hyperparameters such as the learning rate or batch size.
>
> In our discussion with Reviewer J68Z it may not be that clear that we tunned training hyperparameters rather than model hyperparameters. We hope that the above clarification is sufficient to address your concerns and that you will re-evaluate your score decrease.

---

> ### Author Response · Authors · 2022-11-23
> **Response to Reviewer**
>
> > After reading the other reviews, and the authors' responses to them, I am going to reduce my score from 10 to 8, as they used the test set to set hyperparameters. While they claim that the hyperparameters were not particularly impactful on the results, this is a clearly big mistake in terms of ML practice and should be avoided at all costs. Nevertheless, I believe their results still hold (as they used pretty standard hyperparameters, apparently), and the hierarchical nature of their model clearly makes a difference, and is worth publishing.
>
> It is indeed the case that tuning the hyperpameters using the test set may be a mistake. Our wording in our response to Reviewer J68Z was not correct.  Please read the following explanation for a detailed clarification.
>
> Whether or not it is acceptable to use the test set for hyperparameter tuning boils down to what types of hyperparameters you are tunning as well as how the tunning is embedded in the overall training process. Following the simple example [here](https://www.geeksforgeeks.org/svm-hyperparameter-tuning-using-gridsearchcv-ml/), suppose we are training an SVM clasifier using GridSearch for the hyperpameters. The correct way to do it is to train a different model for all combinations of hyperparameters and then use the validation set to pick the best hyperparameters. The wrong way would be to use the validation metric as a training signal. In the later case, we end up training on the validation set. There are two distinct characteristics in the above case:
>
> 1. The dimension of the hyper-parameter space is comparable to the dimension of the model parameter space. There is ~20 model parameters and 2 hyperparameters.
> 2. The hyperparameters are essentially model hyperparameters, i.e., they appear directly in the model definition. This means that different values can lead to vastly different train/validation performance.
>
> All of the above do not apply to our case for the following reasons:
>
> 1. We did not use any metric evaluated in the test set as a signal to guide the model training. What we did is that we picked some inital, reasonable values for the learning rate, batch size and number of epochs, trained on the train set and evaluated on the test set. Then, we varied those hyperparameters as part of our practical expiramentation in order to see how the model performs.
> 2. Our hyperparameter space is really small compared to the model parameter space. We only "tuned" 3 hyperparameters whereas our overall model has parameters in the range of 10^5-^10^6. Even if we were to selectivelly and intentionally overfit 3 parameters, one could hardly say that the model overfits given that a few parameters have little to no impact on the performance of such a complex model.
> 3. Lastly, and most importantly, the hyperparameters that we tunned were training hyperparameters, not model hyperparameters. This means that by varying our hyperparameters we expect (and observed) that the train/test performance will vary in a similar fashion. For example, by increasing the learning rate **both** the train and test loss converged faster but risked overshooting in some runs. By increasing the batch size, the time per epoch was lower but we required higher number of epochs to converge. The purpose of us tunning these hyperparameters was to study those trade offs and **not** to cherry-pick the hyperparameters that will give us the best results. You cannot really overfit to training hyperparameters such as the learning rate or batch size.
>
> In our discussion with Reviewer J68Z it may not be that clear that we tunned training hyperparameters rather than model hyperparameters. We hope that the above clarification is sufficient to address your concerns and that you will re-evaluate your score decrease.

---

### Decision · Program_Chairs · 2023-01-20

**Decision:**

Reject

**Justification For Why Not Higher Score:**

The data leakage concern was not properly addressed and hence I do not believe this paper should be accepted in its current form.

**Justification For Why Not Lower Score:**

NA

**Metareview: Summary, Strengths And Weaknesses:**

The paper presents a novel method for decoding images from FMRI, using a hierarchical VAE. The paper uses the hierarchical bias of visual processing together with the hierarchy of the VAE to achieve good reconstruction results, both quantitatively and qualitatively. However, two reviewers have raised important problems with the method, including data leakage and hence concerns about the generalisability of the method to new images. Overall, the reviewers believe that the improvements shown in this paper come from using a better generative model rather than a fundamental new contribution of the authors, and it is not clear whether this work will be of value to either the ML or neuroscience communities in its current form.